

# Invertible networks or partons to detector and back again

**Marco Bellagente[1], Anja Butter[1\*], Gregor Kasieczka[3], Tilman Plehn[1],**
**Armand Rousselot[1,2], Ramon Winterhalder[1], Lynton Ardizzone[2] and Ullrich Köthe[2]**

**1** Institut für Theoretische Physik, Universität Heidelberg, Germany
**2** Heidelberg Collaboratory for Image Processing, Universität Heidelberg, Germany
**3** Institut für Experimentalphysik, Universität Hamburg, Germany

\* butter@thphys.uni-heidelberg.de

## Abstract

For simulations where the forward and the inverse directions have a physics meaning, invertible neural networks are especially useful. A conditional INN can invert a detector simulation in terms of high-level observables, specifically for ZW production at the LHC. It allows for a per-event statistical interpretation. Next, we allow for a variable number of QCD jets. We unfold detector effects and QCD radiation to a pre-defined hard process, again with a per-event probabilistic interpretation over parton-level phase space.


**Content**

# 1   Introduction

The unique feature of LHC physics from a data science perspective is the comparison of vast amounts of data with predictions based on first principles. This modular prediction starts with the Lagrangian describing the hard scattering, then adds perturbative QCD providing precision predictions, resummed QCD describing parton showers and fragmentation, hadronization, and finally a full detector simulation [1]. In this so-defined forward direction all simulation modules are based on Monte Carlo techniques, and in the ideal world we would just compare measured and simulated events and draw conclusions about the hard process. This hard process is where we expect to learn about new aspects of fundamental physics, for instance dark matter, extended gauge groups, or additional Higgs bosons.

Because our simulation chain works only in one direction, the typical LHC analysis starts with a new, theory-inspired hypothesis encoded in a Lagrangian as new particles and couplings. For every point in the new physics parameter space we simulate events, compare them to the measured data using likelihood methods, and discard the new physics hypothesis. This approach is inefficient for a variety of reasons:

1. The best way to compare two hypotheses is the log-likelihood ratio based on new physics and Standard Model predictions for the hard process. Using this ratio in the analysis is the idea behind the matrix element method [2–7], but usually this information is not available [8].

2. New physics hypotheses have free model parameters like masses or couplings, even if an analysis happens to be independent of them. If the predicted event rates follow a simple scaling, like for a truncated effective theory, this is simple, but usually we need to simulate events for each point in model space.

3. There is a limit in electroweak or QCD precision to which we can reasonably include predictions in our standard simulation tools. Beyond this limit we can, for instance, only compute a limited set of kinematic distributions, which excludes these precision prediction from standard analyses.

4. Without a major effort it is impossible for model builders to derive competitive limits on a new model by recasting an existing analysis.

All these shortcomings point into the same direction: we need to invert the simulation chain, apply this inversion to the measured data, and compare hypotheses at the level of the hard scattering. For hadronization and fragmentation an approximate inversion is standard in that we always apply jet algorithms to extract simple parton properties from the complex QCD jets. For the detector simulation either at the level of particles or at the level of jets this problem is usually referred to as detector unfolding. For instance in top physics we also unfold kinematic information to the level of the decaying top quarks, assuming that the top decays are correctly described by the standard model [9,10]. Going beyond detector effects we know what for many analyses QCD jet radiation adds little to our new physics search. This is certainly true whenever soft and collinear radiation can be simulated by spin-averaged parton showers depending only logaritmically on the global energy scale of the hard process. In that case we should also be able to also unfold QCD jet radiation as the last simulation step. This is the final goal of our paper.

Technically, we propose to use invertible networks (INNs) [11–13] to invert part of the LHC simulation chain. This application builds on a long list of one-directional applications of generative or similar networks to LHC simulations, including phase space integration [14,15], amplitudes [16,17], event generation [18–22], event subtraction [23], detector simulations [24–

32], parton showers [33–36], or searches for physics beyond the Standard Model [37]. INNs are an alternative class of generative networks, based on normalizing flows [38–41]. In particle physics such normalizing flow networks have proven useful for instance in phase space generation [42], linking integration with generation [43, 44], or anomaly detection [45].

Our INN study on unfolding detector-level events [46] to the hard scattering builds on similar attempts with a standard GAN [47] and a fully conditional GAN analysis [48]. In Sec. 3 we show how the bijective structure of the INN makes their training especially stable. If we add sufficiently many random numbers to the INN we can start generating probability distributions in the parton-level phase space. The conditional INN (cINN) [49,50] adds even more sampling elements to the generation of unfolded configurations. For arbitrary kinematic distributions we can test the calibration of this generative network output using truth information and find that unlike GANs the cINN lives up to its generative promise: for a single detector-level event the cINN generates probability distributions in the multi-dimensional parton-level phase space.

Next, we show in Sec. 4 how the inversion can link two phase spaces with different dimension. This allows us to unfold based on a model with a variable number of final state particles at the detector level and is crucial to include higher-order perturbative corrections. We show how the cINN can account for jet radiation and unfolds it together with the detector effects. In other words, the network distinguishes between jets from the hard process and jets from QCD radiation and it also unfolds the kinematic modifications from initial state radiation, to provide probability distributions in the parton-level phase space of a hard process.

We note that our examples only cover analyses where subjet information factorizes from the hard process, for instance in terms of (mis-)tagging efficiencies. For analyses going beyond this level, like searches for long-lived particles, we need to skip the jet algorithm stage and instead include the full calorimeter and tracking information. In principle and assuming the availability of a proper detector simulations our ideas might still work for these applications, but for the time being we ignore these complications.

# 2 Unfolding basics

Unfolding particle physics events is a classic example for an inverse problem [51–53]. In the limit where detector effects can be described by Gaussian noise, it is similar to unblurring images. However, actual detector effects depend on the individual objects, the global energy deposition per event, and the proximity of objects, which means they are much more complicated than Gaussian noise. The situation gets more complicated when we add effects like QCD jet radiation, where the radiation pattern depends for instance on the quantum numbers of the incoming partons and on the energy scale of the hard process.

What we do know is that we can describe the measurement of phase space detector-level distributions $d\sigma/dx_d$ as a random process, just as the detector effects or jet radiation can be simulated by a set of random numbers describing a Markov process. This means that also the inversion or extraction of the parton-level distribution $d\sigma/dx_p$ is a statistical problem.

## 2.1 Binned toy model and locality

As a one-dimensional toy example we can look at a binned (parton-level) distribution $\sigma_j^{(p)}$ which gets transformed into another binned (detector-level) distribution $\sigma_j^{(d)}$ by the kernel or response function $g_{ij}$,

$$\sigma_i^{(d)} = \sum_{j=1}^{N} g_{ij} \sigma_j^{(p)} \,. \tag{1}$$

We can postulate the existence of an inversion with the kernel $\bar{g}$ through the relation

$$\sigma_k^{(p)} = \sum_{i=1}^N \bar{g}_{ki}\sigma_i^{(d)} = \sum_{j=1}^N \left(\sum_{i=1}^N \bar{g}_{ki}g_{ij}\right)\sigma_j^{(p)} \quad \text{with} \quad \sum_{i=1}^N \bar{g}_{ki}g_{ij} = \delta_{kj}. \tag{2}$$

If we assume that we know the $N^2$ entries of the kernel $g$, this form gives us the $N^2$ conditions to compute its inverse $\bar{g}$. We illustrate this one-dimensional binned case with a semi-realistic smearing matrix

$$g = \begin{pmatrix} 1-x & x & 0 \\ x & 1-2x & x \\ 0 & x & 1-x \end{pmatrix}. \tag{3}$$

We illustrate the smearing pattern with two input vectors, keeping in mind that in an unfolding problem we typically only have one kinematic distribution to determine the inverse matrix $\bar{g}$,

$$\sigma^{(p)} = n\begin{pmatrix}1\\1\\1\end{pmatrix} \quad \Rightarrow \quad \sigma^{(d)} = \sigma^{(p)},$$

$$\sigma^{(p)} = \begin{pmatrix}1\\n\\0\end{pmatrix} \quad \Rightarrow \quad \sigma^{(d)} = \sigma^{(p)} + x\begin{pmatrix}n-1\\-2n+1\\n\end{pmatrix}. \tag{4}$$

The first example shows how for a symmetric smearing matrix a flat distribution removes all information about the detector effects. This implies that we might end up with a choice of reference process and phase space such that we cannot extract the detector effects from the available data. The second example illustrates that for bin migration from a dominant peak the information from the original $\sigma^{(p)}$ gets overwhelmed easily. We can also compute the inverse of the smearing matrix in Eq.(3) and find

$$\bar{g} \approx \frac{1}{1-4x}\begin{pmatrix} 1-3x & -x & x^2 \\ -x & 1-2x & -x \\ x^2 & -x & 1-3x \end{pmatrix}, \tag{5}$$

where we neglect the sub-leading $x^2$-terms whenever there is a linear term as well. The unfolding matrix extends beyond the nearest neighbor bins, which means that local detector effects lead to a global unfolding matrix and unfolding only works well if we understand our entire data set. The reliance on useful kinematic distributions and the global dependence of the unfolding define the main challenges once we attempt to unfold the full phase space of an LHC process.

## 2.2 Bayes' theorem and model dependence

Over the continuous phase space a detector simulation can be written as

$$\frac{\mathrm{d}\sigma}{\mathrm{d}x_d} = \int \mathrm{d}x_p \; g(x_d, x_p) \frac{\mathrm{d}\sigma}{\mathrm{d}x_p}, \tag{6}$$

where $x_d$ is a kinematic variable at detector level, $x_p$ the same variable at parton level, and $g$ a kernel or transfer function which links these two arguments. We ignore efficiency factors for

now, because they can be absorbed into the parton-level rate. To invert the detector simulation we define a second transfer function $\bar{g}$ such that [54–56]

$$\frac{d\sigma}{dx_p} = \int dx_d \, \bar{g}(x_p, x_d) \frac{d\sigma}{dx_d} = \int dx'_p \frac{d\sigma}{dx'_p} \int dx_d \, \bar{g}(x_p, x_d) g(x_d, x'_p) \,. \tag{7}$$

This inversion is fulfilled if we construct the inverse $\bar{g}$ of $g$ defined by

$$\int dx_d \, \bar{g}(x_p, x_d) g(x_d, x'_p) = \delta(x_p - x'_p) \,, \tag{8}$$

all in complete analogy to the binned form above. The symmetric form of Eq.(6) and Eq.(7) indicates that $g$ and $\bar{g}$ are both defined as distributions. In the $g$-direction we use Monte Carlo simulation and sample in $x_p$, while $\bar{g}$ needs to be sampled in $g(x_p)$ or $x_d$. In both directions this statistical nature implies that we should only attempt to unfold sufficiently large event samples.

The above definitions can be linked to Bayes' theorem if we identify the kernels with probabilities. We now look at $\bar{g}(x_d|x_p)$ in the slightly modified notation as the probability of observing $x_d$ given the model prediction $x_p$ and $g(x_p|x_d)$ gives the probability of the model $x_p$ being true given the observation $x_d$ [57, 58]. In this language Eq.(6) and (7) describe conditional probabilities, and we can write something analogous to Bayes' theorem,

$$\bar{g}(x_p|x_d) \frac{d\sigma}{dx_d} \sim g(x_d|x_p) \frac{d\sigma}{dx_p} \,. \tag{9}$$

In this form $\bar{g}(x_p|x_d)$ is the posterior, $g(x_d|x_p)$ as a function of $x_p$ is the likelihood, $d\sigma/dx_p$ is the prior, and the model evidence $d\sigma/dx_d$ fixes the normalization of the posterior. From standard Bayesian analyses we know two things: (i) the posterior will in general depend on the prior, in our case the kinematics of the underlying particle physics process or model; (ii) when analyzing high-dimensional spaces the prior dependence will vanish when the likelihood develops a narrow global maximum.

If the posterior $\bar{g}(x_p|x_d)$ in general depends on the model $d\sigma/dx_p$, then Eq.(7) does not look useful. On the other hand, Bayesian statistics is based on the assumption that the prior dependence of the posterior defines an iterative process where we start from a very general prior and enter likelihood information step by step to finally converge on the posterior. The same approach can define a kinematic unfolding algorithm [59]. We will not discuss these methods further, but come back to this model dependence throughout our paper.

## 2.3 Reference process $pp \to ZW$

To provide a quantitative estimate of unfolding with an invertible neural networks we use the same example process as in Ref. [48],

$$pp \to ZW^{\pm} \to (\ell^- \ell^+) (jj) \,, \tag{10}$$

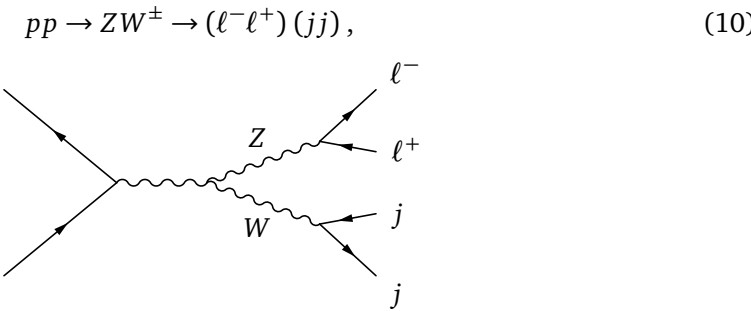

Figure 1: Sample Feynman diagram contributing to $ZW$ production, with intermediate on-shell particles labeled.

One of the contributing Feynman diagrams is shown in Fig. 1. With jets and leptons in the final state we can test the stability of the unfolding network for small and for large detector effects. We generate the $ZW$ events using MADGRAPH5 [60] without any generation cuts and then simulate parton showering with PYTHIA8 [61] and the detector effects with DELPHES [62] using the standard ATLAS card. For jet clustering we use the anti-$k_T$ algorithm [63] with $R = 0.6$ implemented in FASTJET [64]. All jets are required to have

$$p_{T,j} > 25 \text{ GeV} \qquad \text{and} \qquad |\eta_j| < 2.5 \,. \tag{11}$$

For the hadronically decaying $W$-boson the limited calorimeter resolution will completely dominate over the parton-level Breit-Wigner distribution. After applying the cuts we have 320k events which we split into 90% training and 10% test data.

In a first step, corresponding to Ref. [48] we are only interested in inverting these detector effects. These results are shown in Sec. 3. For the simulation this implies that we switch off initial state radiation as well as underlying event and pile-up effects and require exactly two jets and a pair of same-flavor opposite-sign leptons. The jets and corresponding partons are separately ordered by $p_T$. The detector and parton level leptons are assigned by charge. This gives us two samples matched event by event, one at the parton level ($x_p$) and one including detector effects ($x_d$). Each of them is given as an unweighted set of four 4-vectors. These 4-vectors can be simplified if we assume all external particles at the parton level to be on-shell. Obviously, this method can be easily adapted to weighted events.

In a second step we include initial state radiation and allow for additional jets in Sec. 4. We still require a pair of same-flavor opposite-sign leptons and at least two jets in agreement with the condition in Eq.(11). The four jets with highest $p_T$ are then used as input to the network, ordered by $p_T$. Events with less than 4 jets are zero-padded. This second data set is only used for the conditional INN.

## 3 Unfolding detector effects

We introduce the conditional INN in two steps, starting with the non-conditional, standard setup. The construction of the INN we use in our analysis combines two goals [11]:

1. the mapping from input to output is invertible and the Jacobians for both directions are tractable;

2. both directions can be evaluated efficiently. This second property goes beyond some other implementations of normalizing flow [38, 40].

While the final aim is not actually to evaluate our INN in both directions, we will see that these networks can be extremely useful to invert a Markov process like detector smearing. Their bi-directional training makes them especially stable.

In Sec. 3.3 we will show how the conditional INN retains a proper statistical notion of the inversion to parton level phase space. This avoids a major weakness of standard unfolding methods, namely that they only work on large enough event samples condensed to one-dimensional or two-dimensional kinematic distributions. This could be a missing transverse energy distribution in mono-jet searches or the rapidities and transverse momenta in top pair production. To avoid systematics or biases in the full phase space coverage required by the matrix element method, the unfolding needs to construct probability distributions in parton-level phase space, including small numbers of events in tails of kinematic distributions.

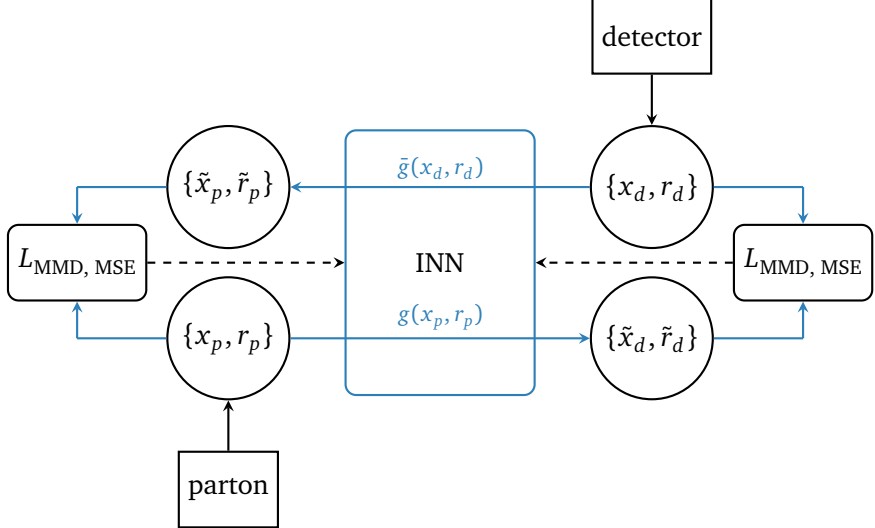

Figure 2: Structure of INN. The $\{x_{d,p}\}$ denote detector-level and parton-level events, $\{r_{d,p}\}$ are random numbers to match the phase space dimensionality. A tilde indicates the INN generation.

## 3.1 Naive INN

While it is clear from our discussion in Ref. [48] that a standard INN will not serve our purpose, we still describe it in some detail before we extend it to a conditional network. Following the conventions of our GAN analysis and in analogy to Eqs.(6) to (8) we define the network input as a vector of hard process information $x_p \in R^{D_p}$ and the output at detector level via the vector $x_d \in \mathbb{R}^{D_d}$. If the dimensionalities of the spaces are such that $D_p < D_d$ we add a noise vector $r$ with dimension $D_d - D_p$ to define the bijective, invertible transformation,

$$\begin{pmatrix} x_p \\ r \end{pmatrix} \xleftrightarrow[\leftarrow \text{ unfolding} : \bar{g}]{\text{PYTHIA,DELPHES} : g \rightarrow} x_d \ . \tag{12}$$

A correctly trained network $g$ with the parameters $\theta$ then reproduces $x_d$ from the combination $x_p$ and $r$. Its inverse $\bar{g}$ instead reproduces the combination of $x_p$ and $r$ from $x_d$.

The defining feature of the INN illustrated in Fig. 2 is that it learns both directions of the bijective mapping in parallel and encodes them into one network. Such a simultaneous training of both directions is guaranteed by the building blocks of the network, the invertible coupling layers [12,65]. For notational purposes we ignore the random numbers in Eq.(12) and assume that this layer links an input vector $x_p$ to an output vector $x_d$ after splitting both of them in halves, $x_{p,i}$ and $x_{d,i}$ for $i = 1, 2$. The relation between input and output is given by a sub-network, which encodes arbitrary functions $s_{1,2}$ and $t_{1,2}$. Using an element-wise multiplication $\odot$ and sum one could for instance define an output $x_{d,1}(x_p) = x_{p,1} \odot s_2(x_{p,2}) + t_2(x_{p,2})$. In order to avoid numerical instabilities caused by the division with $s(x)$ in the inverse direction, we include an exponential to obtain

$$\begin{pmatrix} x_{d,1} \\ x_{d,2} \end{pmatrix} = \begin{pmatrix} x_{p,1} \odot e^{s_2(x_{p,2})} + t_2(x_{p,2}) \\ x_{p,2} \odot e^{s_1(x_{d,1})} + t_1(x_{d,1}) \end{pmatrix} \iff \begin{pmatrix} x_{p,1} \\ x_{p,2} \end{pmatrix} = \begin{pmatrix} (x_{d,1} - t_2(x_{p,2})) \odot e^{-s_2(x_{p,2})} \\ (x_{d,2} - t_1(x_{d,1})) \odot e^{-s_1(x_{d,1})} \end{pmatrix} . \tag{13}$$

By construction, this inversion works independent of the form of $s$ and $t$. If we write the coupling block function as $g(x_p) \sim x_d$, again omitting the random numbers $r$, the Jacobian of

the network function has a triangular form

$$\frac{\partial g(x_p)}{\partial x_p} = \begin{pmatrix} \text{diag } e^{s_2(x_{p,2})} & \text{finite} \\ 0 & \text{diag } e^{s_1(x_{d,1})} \end{pmatrix}, \tag{14}$$

so its determinant is easy to compute. Such coupling layer transformations define a so-called normalizing flow, when we view it as transforming an initial probability density into a very general form of probability density through a series of invertible steps. We can relate the two probability densities as long as the Jacobians of the individual layers can be efficiently calculated.

Since the first use of the invertible coupling layer, much effort has gone into improving its efficiency. The All-in-One (AIO) coupling layer includes two features, introduced by Ref. [12] and Ref. [13]. The first modification replaces the transformation of $x_{p,2}$ by a permutation of the output of each layer. Due to the permutation each component still gets modified after passing through several layers. The second modification includes a global affine transformation to include a global bias and linear scaling that maps $x \to sx + b$. Finally, we apply a bijective soft clamping after the exponential function in Eq.(13) to prevent instabilities from diverging outputs.

The INN in our simplified example combines three contributions to the loss function. First, it tests if in the DELPHES direction of Eq.(12) we indeed find $g(x_p) = x_d$ via the mean squared error (MSE) function. While this is theoretically sufficient to obtain the inverse function, also testing the inverse direction $\bar{g}(x_d) = x_p$ greatly improves the efficiency and stability of the training. Third, to resolve special sharp features like the invariant mass of intermediate particles we use the maximum mean discrepancy (MMD) as a distance measure between the generated and real distribution of these features.

Because we will also use the MMD in another function function [21] we review it briefly. An MMD loss allows us to compare any pre-defined distribution. For a relativistic phase space a critical narrow phase space feature is the invariant mass of intermediate particles. We can force the network to consider this one-dimensional distribution of the 4-vectors $x_p$ for batches of parton-level and detector-level events,

$$\text{MMD} = \left[ \langle k(x, x') \rangle_{x,x' \sim P_p} + \langle k(y, y') \rangle_{y,y' \sim P_d} - 2 \langle k(x, y) \rangle_{x \sim P_p, y \sim P_d} \right]^{1/2}. \tag{15}$$

In Refs. [21] and [48] we compare common choices, like Gaussian or Breit-Wigner kernels

$$k_{\text{Gauss}}(x, y) = \exp \frac{-(x-y)^2}{2\sigma^2} \qquad \text{or} \qquad k_{\text{BW}}(x, y) = \frac{\sigma^2}{(x-y)^2 + \sigma^2}, \tag{16}$$

with a fixed or variable width $\sigma$ [48]. Inside the INN architecture the Breit-Wigner kernel is the best choice to analyze the distribution of the random numbers as part of the loss function [11].

We now use the INN network to map parton-level events to detector-level events or vice-versa. In a statistical analysis we then use standard kinematic distributions and compare the respective truth and INN-inverted shapes for both directions. The left panels of Fig. 3 shows the transverse momentum distributions of the two jets and their invariant mass for both directions of the INN. The truth events at parton level and at detector level are marked as dashed lines. Starting from each of the truth events we can apply the INN describing the detector effects as $x_d = g(x_p)$ or unfolding the detector effects as $x_p = \bar{g}(x_d)$ in Eq.(12). The corresponding solid lines have to be compared to the dotted truth lines, where we need to keep in mind that at the parton level the relevant objects are quarks while at the detector level they are jets.

For the leading jet the truth and INNed detector-level agree very well, while for the second jet the naive INN fails to capture the hard cut imposed by the jet definition. For the invariant mass we find that the smearing due to the detector effects is reproduced well with some small

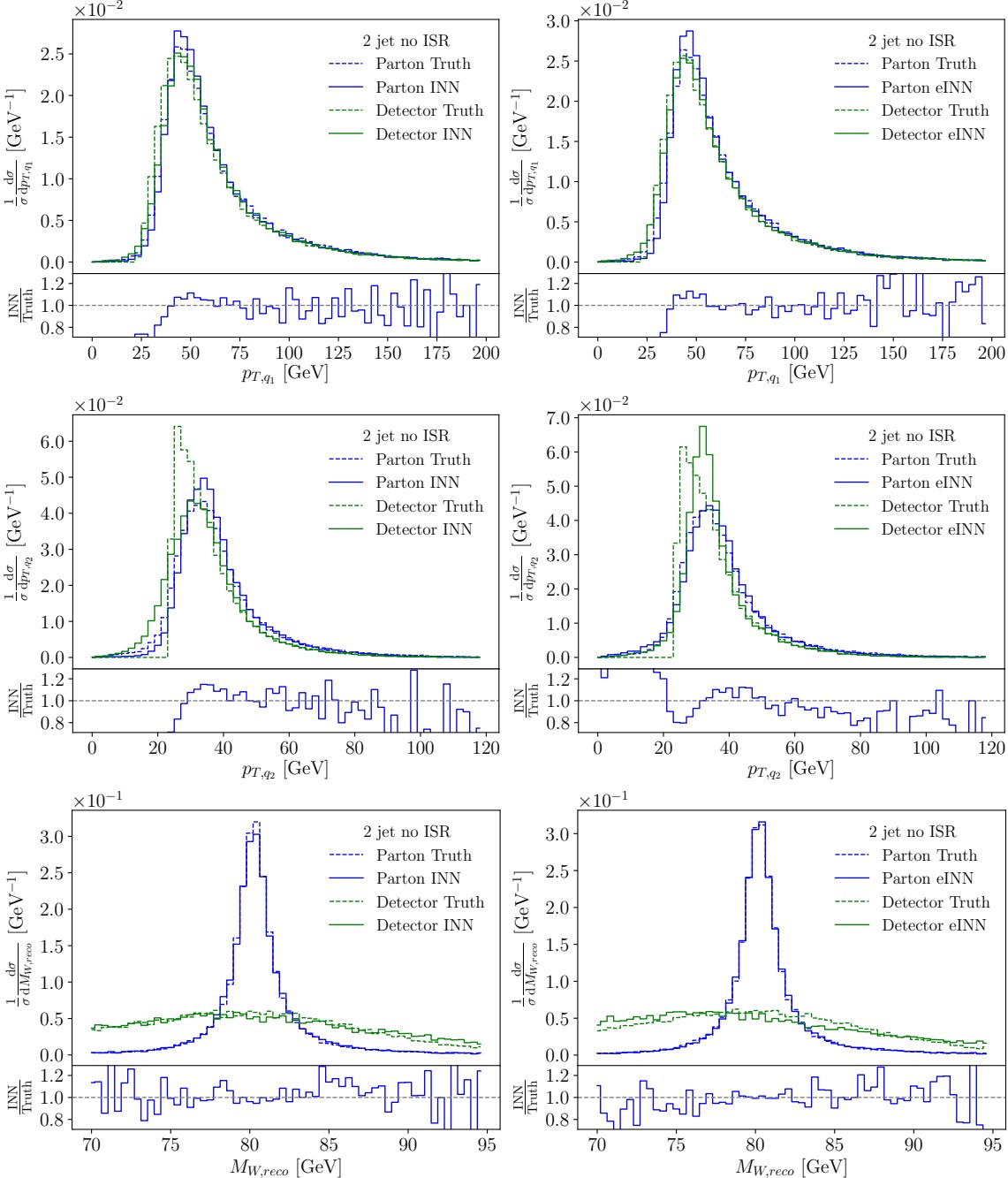

Figure 3: INNed $p_{T,q}$ and $M_{W,\mathrm{reco}}$ distributions from a naive INN (left) and the noise-extended eINN (right). In green we compare the detector-level truth to INNed events transformed from parton level. In blue we compare the parton-level truth to INNed events transformed from detector level. The secondary panels show the ratio of INNed events over parton-level truth. More distributions can be found in the pdf files submitted to the arXiv.

deviations in the tails. In the unfolding direction both $p_T$ distributions follow the parton level truth. The only difference is a systematic lack of events in the tail for the second quark. This is especially visible in the ratio of the INN-unfolded events and the parton-level truth, indicating that also at small $p_T$ the network does not fill the phase space sufficiently. Combining both directions we see that in forward direction the INN produces a too broad $p_T$-distribution, the

unfolding direction of the INN produces a too narrow distribution. The conceptual advantage of the INN actually implies a disadvantage for the inversion of particular difficult features. Finally, the invariant mass of the $W$ is reproduced perfectly without any systematic deviation.

## 3.2 Noise-extended INN

While our simplified example in the previous section shows some serious promise of INNs, it fails to incorporate key aspects of the physical process. First of all, the number of degrees of freedom is not actually the same at parton level and at detector level. External partons are on their mass shell, while jets come with a range of jet masses. This mismatch becomes crucial when we include missing transverse momentum in the signature. We generally need fewer parameters to describe the partonic scattering than the detector-level process. For a fixed set of parton-level momenta we usually smear each momentum component to simulate the detector measurement. These additional degrees of freedom are of stochastic nature, so adding Gaussian random variable on the parton side of the INN could be a first step to address this problem.

To also account for potentially unobservable degrees of freedom at the parton level we extend each side of the INN by a random number vector. The mapping in Eq.(12) now includes two random number vectors with dimensions $D_{r_d} = D_p$ and $D_{r_p} = D_d$,

$$
\begin{pmatrix} x_p \\ r_p \end{pmatrix} \xleftarrow[\leftarrow \text{unfolding:}\bar{g}]{\text{PYTHIA,DELPHES:}g\rightarrow} \begin{pmatrix} x_d \\ r_d \end{pmatrix} . \tag{17}
$$

In addition, a pure MSE loss can not capture the fact that the additional noise generates a distribution of detector-level events given fixed parton momenta. It would just predict of a mean value of this distribution and minimize the effect of the noise. A better solution is an MMD loss for each degree of freedom in the event and the masses of intermediate particles, as well as the Gaussian random variables. On the side of the random numbers this MMD loss ensures that they really only encode noise. Again it is beneficial for the training to use the inverse direction and apply additional MMD losses to the parton level events as well as the corresponding Gaussian inputs. Finally we add a weak MSE loss on the four vectors of each side to stabilize the training.

In the right panels of Fig. 3 we show results for this noise-extended INN (eINN). The generated distributions are similar to the naive INN case and match the truth at the parton level. A notable difference appears in the second jet, the weak spot of the naive INN. The additional random numbers and MMDs provide more freedom to generate the peak in the forward direction and also improve the unfolding in the low-$p_T$ and high-$p_T$ regimes.

Aside from the better modeling, the noise extension allows for a statistic interpretation of the generated distributions and a test of the integrity of the INN-inverted distributions. In the left panel of Fig. 4 we illustrate the goal of the statistical treatment: we start from a single event at the detector level and generate a set of unfolded events. For each of them we evaluate for instance $p_{T,q_1}$. Already in this illustration we see that the GAN output is lacking a statistical behavior at the level of individual events, while the noise-extended eINN returns a reasonable distribution of unfolded events.

To see if the width of this INN output is correct we take 1500 parton-level and detector-level event pairs and unfold each event 60 times, sampling over the random variables. This gives us 1500 combinations like the one shown in the left panel of Fig. 4: a single parton-level truth configuration and a distribution of the INNed configuration. To see if the central value and the width of the INNed distribution can be interpreted statistically as a posterior probability distribution in parton phase space we analyse where the truth lies within the INN

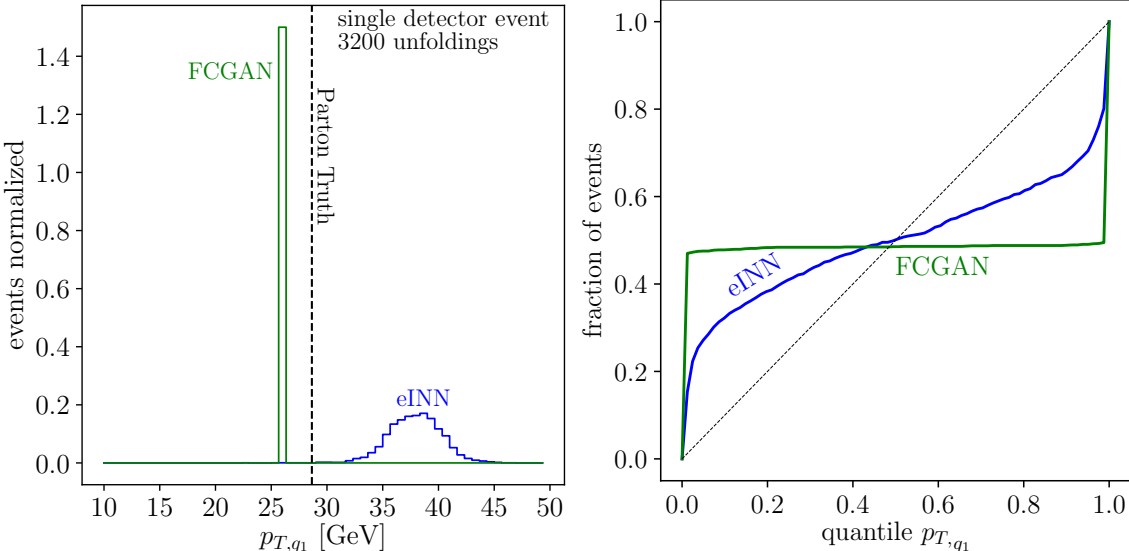

Figure 4: Left: illustration of the statistical interpretation of unfolded events for one event. Right: calibration curves for $p_{T,q_1}$ extracted from the FCGAN and the noise-extended eINN.

distribution for each of the 1500 events. For a correctly calibrated curve we start for instance from the left of the kinematic distribution and expect 10% of the 1500 events in the 10% quantile of the respective probability distribution, 20% of events in the 20% quantile, etc. The corresponding calibration curves for the noise-extended eINN are shown in the right panel of Fig. 4. While they indicate that we can attempt a statistical interpretation of the INN unfolding, the calibration is not (yet) perfect. A steep rise for the lower quantile indicates that too many events end up in the first 10% quantile. In other words, the distributions we obtain by sampling over the Gaussian noise for each event are too narrow.

While our noise-extended eINN takes several steps in the right direction, it still faces major challenges: the combination of many different loss functions is sensitive to their relative weights; the balance between MSE and MMD on event constituents has to be calibrated carefully to generate reasonable quantile distributions; when we want to extend the INN to include more detector-level information we have to include an equally large number of random variable on the parton level which makes the training very inefficient. This leads us again [48] to adopt a conditional setup.

## 3.3 Conditional INN

If a distribution of parton-level events can be described by $n$ degrees of freedom, we should be able to use normalizing flows or an INN to map a $n$-dimensional random number vector onto parton-level 4-momenta. To capture the information from the detector-level events we need to condition the INN on these events [48, 66, 67], so we link the parton-level data $x_p$ to random noise $r$ under the condition of $x_d$. Trained on a given process the network should now be able to generate probability distributions for parton-level configurations given a detector-level event and an unfolding model. We note that the cINN is still invertible in the sense that it includes a bi-directional training from Gaussian random numbers to parton-level events and back. While this bi-directional training does not represent the inversion of a detector simulation anymore, it does stabilize the training by requiring the noise to be Gaussian.

A graphic representation of this conditional INN or cINN is given in Fig. 5. We first process the detector-level data by a small subnet, *i.e.* $x_d \rightarrow f(x_d)$, to optimize its usability for the

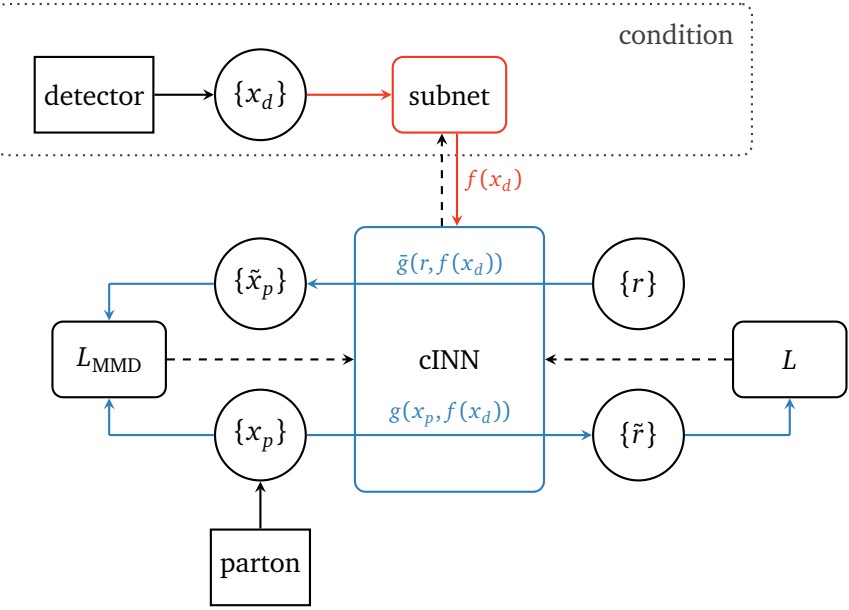

Figure 5: Structure of the conditional INN. The input are random numbers $\{r\}$ while $\{x_{d,p}\}$ denote detector-level and parton-level data. The latent dimension loss $L$ follows Eq.(18), a tilde indicates the INN generation.

cINN [49]. The subnet is trained alongside the cINN and does not need to be reversed or adapted. We choose a shallow and wide architecture of two layers with a width of 1024 internally, because four layers degrade already the conditional information and allow the cINN to ignore it. When a deeper subnet is required we advertize to use an encoder, which is initialized by pre-training it as part of an autoencoder. We apply this technique when using the larger ISR input, where it leads to a more efficient training. After this preprocessing, the detector information is passed to the functions $s_i$ and $t_i$ in Eq.(13), which now depend on the input, the output, and on the fixed condition. Since the invertibility of the network is independent of the values of $s_i$ and $t_i$, the network remains invertible between the parton-level events $\{x_p\}$ and the random variables $\{r\}$. This feature stabilizes the training. The cINN loss function is motivated by the simple argument that for the correct set of network parameters

Table 1: INN and noise-extended eINN setup and hyper-parameters, as implemented in PyTorch (v1.2.0) [68].

| Parameter | INN | eINN |
|---|---|---|
| Blocks | 24 | 24 |
| Layers per block | 2 | 2 |
| Units per layer | 256 | 256 |
| Trainable weights | $\sim 150$k | $\sim 270$k |
| Epochs | 1000 | 1000 |
| Learning rate | $8 \cdot 10^{-4}$ | $8 \cdot 10^{-4}$ |
| Batch size | 512 | 512 |
| Training/testing events | 290k / 30k | 290k / 30k |
| Kernel widths | $\sim 2, 8, 25, 67$ | $\sim 2, 8, 25, 67$ |
| $D_p + D_{r_p}$ | $12 + 4$ | $12 + 16$ |
| $D_d + D_{r_d}$ | $16 + 0$ | $16 + 12$ |
| $\lambda_{\text{MMD}}$ | 0.1 (masses only) | 0.2 |
| $\lambda_{\text{MMD}}$ increase | - | - |



Figure 6: cINNed $p_{T,q}$ and $m_{W,\text{reco}}$ distributions. Training and testing events include exactly two jets. In the left panels we use a data set without ISR, while in the right panels we use the two-jet events in the full data set with ISR. The lower panels give the ratio of cINNed to parton-level truth.

$\theta$ describing $s_i$ and $t_i$ we maximize the (posterior) probability $p(\theta|x_p, x_d)$ or minimize

$$
\begin{aligned}
L &= -\Big\langle \log p(\theta|x_p, x_d) \Big\rangle_{x_p \sim P_p, x_d \sim P_d} \\
&= -\Big\langle \log p(x_p|x_d, \theta) + \log p(\theta|x_d) - \log p(x_p|x_d) \Big\rangle_{x_p \sim P_p, x_d \sim P_d} \\
&= -\Big\langle \log p(x_p|x_d, \theta) \Big\rangle_{x_p \sim P_p, x_d \sim P_d} - \log p(\theta) + \text{const.} \\
&= -\left\langle \log p(g(x_p, x_d)) + \log \left| \frac{\partial g(x_p, x_d)}{\partial x_p} \right| \right\rangle_{x_p \sim P_p, x_d \sim P_d} - \log p(\theta) + \text{const.} ,
\end{aligned}
\tag{18}
$$

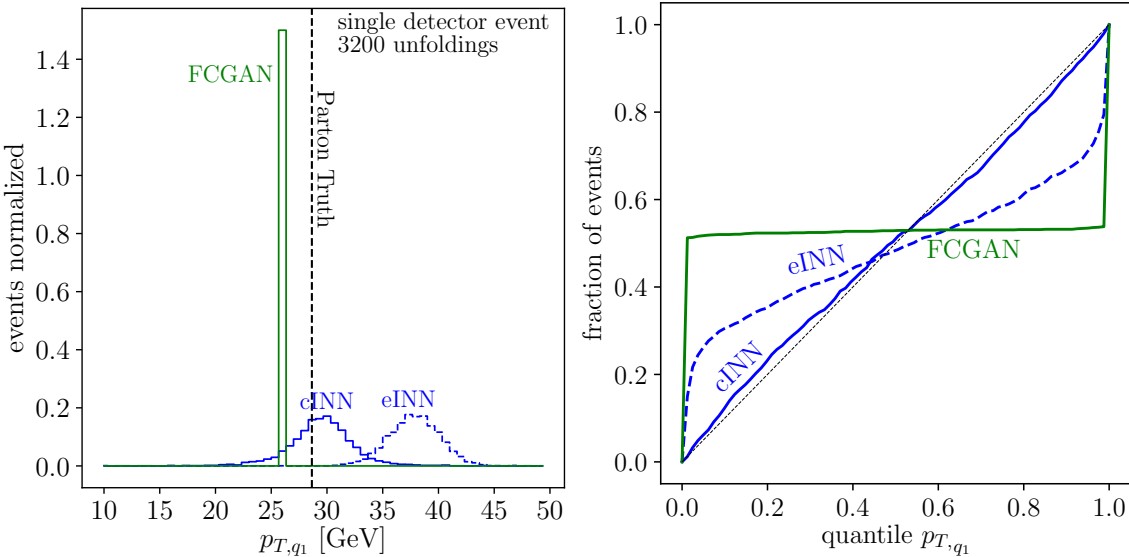

Figure 7: Left: illustration of the statistical interpretation of unfolded events for one event. Right: calibration curves for $p_{T,q_1}$ extracted from the FCGAN and the noise-extended eINN, as shown in Fig. 4, and the cINN.

where we first use Bayes' theorem, then ignore all terms irrelevant for the minimization, and finally apply a simple coordinate transformation for the bijective mapping. The last term is a simple weight regularization, while the first two terms are called the maximum likelihood loss. Since we impose the latent distribution of the random variable $p(g(x_p, x_d))$ to produce a normal distribution centered around zero and with width one, the first term becomes

$$\log p(g(x_p, x_d)) = -\frac{||g(x_p, x_d))||_2^2}{2} \, . \tag{19}$$

The final network setup after tuning of the hyper-paramaeters are liste In Tab. 2. We verified that the network performance is stable under small changes of these parameters.

In the left panels of Fig. 6 we show the unfolding performance of the cINN, trained and tested on the same exclusive 2-jet events as the simpler INNs in Fig. 3. Unlike the naive and the noise-extended INNs we cannot evaluate the cINN in both direction, detector simulation and unfolding, so we focus on the detector unfolding. The agreement between parton-level truth and the INN-unfolded distribution is around 10% for the bulk of the $p_T$ distributions, with the usual larger relative deviations in the tails. An interesting feature is still the cut $p_{T,j} > 20$ GeV at the detector level, because it leads to a slight shift in the peak of the $p_{T,j_2}$ distribution. Finally, the reconstructed invariant $W$-mass and the physical $W$-width agree extremely well with the Monte Carlo truth owing to the MMD loss.

As in Fig. 4 we can interpret the unfolding output for a given detector-level event statistically. First, in the left panel of Fig. 7 we show a single event and how the FCGAN, INN, and cINN output is distributed in parton level phase space[*]. The separation between truth and sampled distributions does not have any significance, but we see that the cINN inherits the beneficial features of the noise-extended eINN. In the right panel of Fig. 7 we again reconstruct the individual probability distribution from the unfolding numerically. We then determine the position of the parton-level truth in its respective probability distribution for the INN and the

---

[*]Throughout this paper we only compare to the FCGAN analysis [48], which we fully control. For standard unfolding methods used by ATLAS and CMS and for the new Omnifold method [46] we refrain from comments which would need to be based on an in-depth comparison.

cINN. We expect a given percentage of the 1500 events to fall into the correct quantile of its respective probability distribution. The corresponding calibration curve for the cINN is added to the right panel of Fig. 7, indicating that without additional calibration the output of the cINN unfolding can be interpreted as a probability distribution in parton-level phase space for a single detector-level event, as always assuming an unfolding model. Instead of the transverse momentum of the harder parton-level quark we could use any other kinematic distribution at parton level. This marks the final step for a statistically interpretable unfolding.

# 4 Unfolding with jet radiation

In the previous chapter we use a simplified data set to explore different possibilities to unfold detector level information with invertible networks. We limit the data to events with exactly two jets, by switching off initial state radiation (ISR). This guarantees that the two jets come from the $W$-decay, so the network does not have to learn this feature. In a realistic QCD environment we do not have that information, because additional QCD jets will be radiated off the initial and final state partons. In this section we demonstrate how we can unfold a sample of events including ISR and hence with a variable number of jets. We know that with very few exceptions [69,70] the radiation of QCD jets does not help us understand the nature of the hard process. In such cases, we would like to interpret a measurement with an appropriately defined hard process, leading to the question if an unfolding network can invert detector effects and QCD jet radiation. Technically, this means inverting jet radiation and kinematic modifications to the hard process as, in our case, done by PYTHIA.

We emphasize that this approach requires us to define a specific hard process with any number of external jets and other features. We can illustrate this choice for two examples. First, a di-tau resonance search typically probes the hard process $pp \to \mu^+\mu^- + X$, where $X$ denotes any number of additional, analysis-irrelevant jets. We invert the corresponding measurements to the partonic process $pp \to \mu^+\mu^-$. A similar mono-jet analysis instead probes the process $pp \to Z'j(j) + X$, where $Z'$ is a dark matter mediator decaying to two invisible dark matter candidate. Depending on the analysis, the relevant process to invert is $pp \to Z'j$ or $pp \to Z'jj$, where a reported missing transverse momentum recoils against one or two hard jets. Because our inversion network in trained on Monte Carlo data, we automatically define the appropriate hard process when generating the training data. This covers any combination of signal and background matrix elements contributing to such a hard process, even non-SM processes to quantify a remaining model dependence. A final caveat — in the hard process we do not include subjet aspects at this stage. As long as subjet information is used for tagging purposes it factorizes from the hard process information and can easily be included in terms of efficiencies. A problem would arise in unfolding or inverting analyses relying on different hard processes, like a fat mono-jet analysis, where the above choice of recoil jets is left to a sub-jet algorithm.

## 4.1 Individual $n$-jet samples

In Sec. 3.3 we have shown that our cINN can unfold detector effects for $ZW$-production at the LHC. The crucial new feature of the cINN is that it provides probability distribution in parton-level phase space for a given detector-level event. The actual unfolding results are illustrated in Fig. 6, focusing on the two critical distribution known from the corresponding FCGAN analysis [48]. The event sample used throughout Sec. 3 includes exactly two partons from a $W$-decay with minimal phase space cuts on the corresponding jets. Strictly speaking, these phase space cuts are not necessary in this simulation. The correct definition of a process

described by perturbative QCD includes a free number of additional jets,

$$pp \to ZW^{\pm} + \text{jets} \to (\ell^- \ell^+)(jj) + \text{jets} .\tag{20}$$

For the additional jets we need to include for instance a $p_T$ cut to regularize the soft and collinear divergences at fixed-order perturbation theory. The proper way of generating events is therefore to allow for any number of additional jets and then cut on the number of hard jets. Since ISR can lead to jets with larger $p_T$ than the $W$-decay jets, an assignment of the hardest jets to hard partons does not work. We simply sort jets and partons by their respective $p_T$ and let the network work out their relations. We limit the number of jets to four because larger jet number appear very rarely and would not give us enough training events.

Combining all jet multiplicities we use 780k events, out of which 530k include exactly two jets, 190k events include three jets and 60k have four or more jets. We split the data into 80% training data and 20% test data to produce the shown plots. For the network input we zero-pad the event-vector for events with less than four jets and add the number of jets as additional information. The training samples are then split by exclusive jet multiplicity, such that the cINN reconstructs the 2-quark parton-level kinematics from two, three, and four jets at the detector level.

As before, we can start with the sample including exactly two jets. The difference to the sample used before is that now one of the $W$-decay jets might not pass the jet $p_T$ condition in Eq.(11), so it will be replaced by an ISR jet in the 2-jet sample. Going back to Fig. 6 we see in the right panel how these events are slightly different from the sample with only decay jet. The main difference is in $p_{T,q_2}$, where the QCD radiation produces significantly more soft jets. Still, the network learns these features, and the unfolding for the sample without ISR and the 2-jet exclusive sample has a similar quality. In Fig. 8 we see the same distributions for the exclusive 3-jet and 4-jet samples. In this case we omit the secondary panels because they are dominated by the statistical uncertainties of the training sample. For these samples the network has to extract the parton-level kinematics with two jets only from up to four jets in the final state. In many cases this corresponds to just ignoring the two softest jets and mapping the two hardest jets on the two $W$-decay quarks, but from the $p_{T,q_2}$ distributions in Fig. 6 we know that this is

Table 2: cINN setup and hyper-parameters, as implemented in PYTORCH (v1.2.0) [68].

| Parameter | cINN no ISR | cINN ISR incl. |
|---|---|---|
| Blocks | 24 | 24 |
| Layers per block | 2 | 3 |
| Units per layer | 256 | 256 |
| Condition/encoder layers | 2 | 8 |
| Units per condition/encoder layer | 1024 | 1024 |
| Condition/encoder output dimension | 256 | 256 |
| Trainable weights | $\sim 2$ M | $\sim 10$ M |
| Encoder pre training epochs | - | 300 |
| Epochs | 1000 | 900 |
| Learning rate | $8 \cdot 10^{-4}$ | $8 \cdot 10^{-4}$ |
| Batch size | 512 | 512 |
| Training/testing events | 290k / 30k | 620k / 160k |
| Kernel widths | $\sim 2, 8, 25, 67$ | $\sim 2, 8, 25, 67$ |
| $D_p$ | 12 | 12 |
| $D_d$ | 16 | 25 |
| $\lambda_{\text{MMD}}$ | 0.5 | 0.04 |
| $\lambda_{\text{MMD}}$ increase | - | 1.6 / 100 epochs |

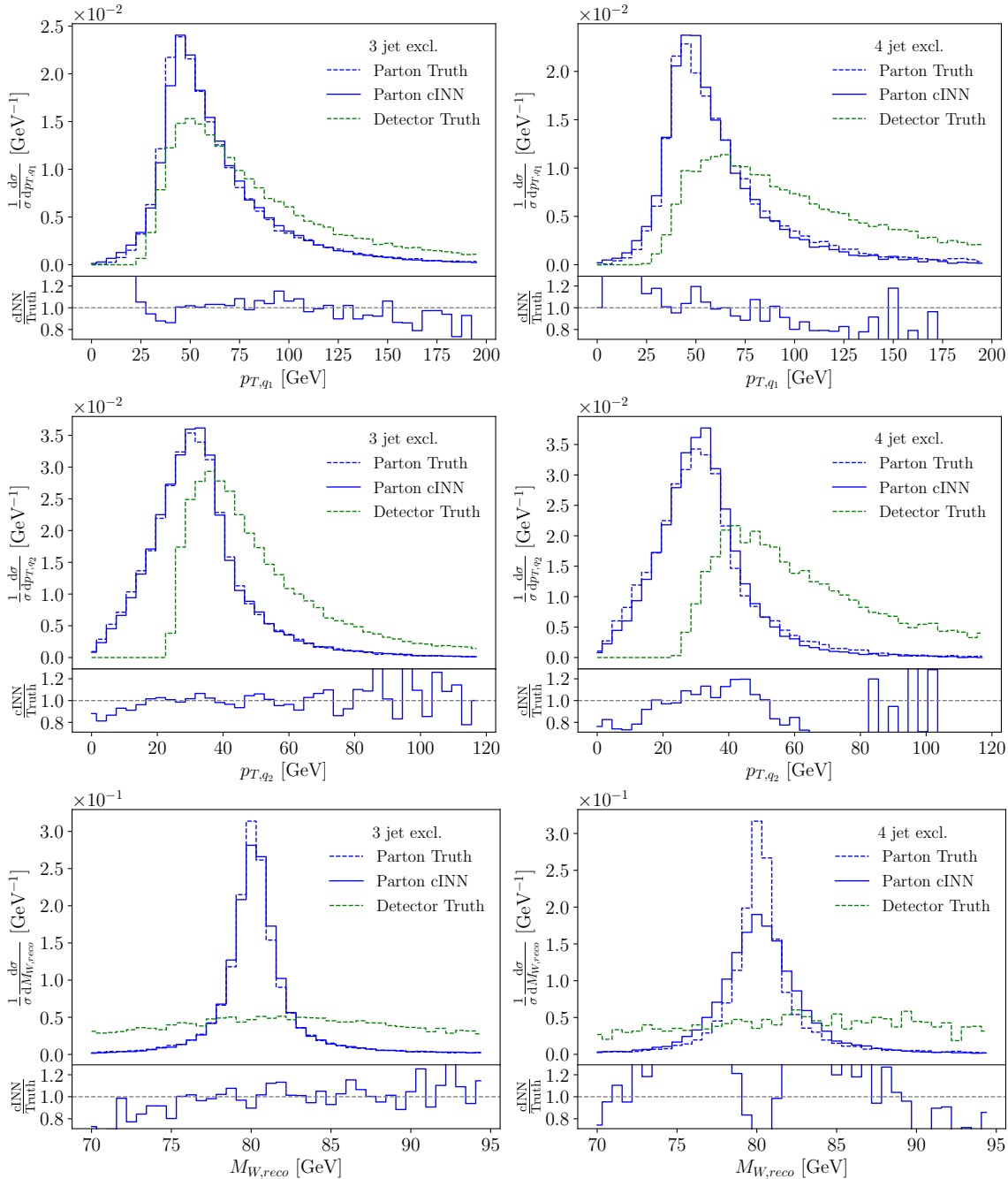

Figure 8: cINNed $p_{T,q}$ and $m_{W,\text{reco}}$ distributions. Training and testing events include exactly three (left) and four (right) jets from the data set including ISR.

not always the correct solution. Especially in the critical $m_{jj}$ peak reconstruction we see that the network feels the challenge, even though the other unfolded distributions look fine.

## 4.2 Combined $n$-jet sample

The obvious final question is if our INN can also reconstruct the hard scattering process with its two $W$-decay quarks from a sample with a variable number of jets. Instead of separate samples as in Sec. 4.1 we now interpret the process in Eq.(20) as jet-inclusive. This means that the hard process includes only the two $W$-decay jets, and all additional jets are understood as jet

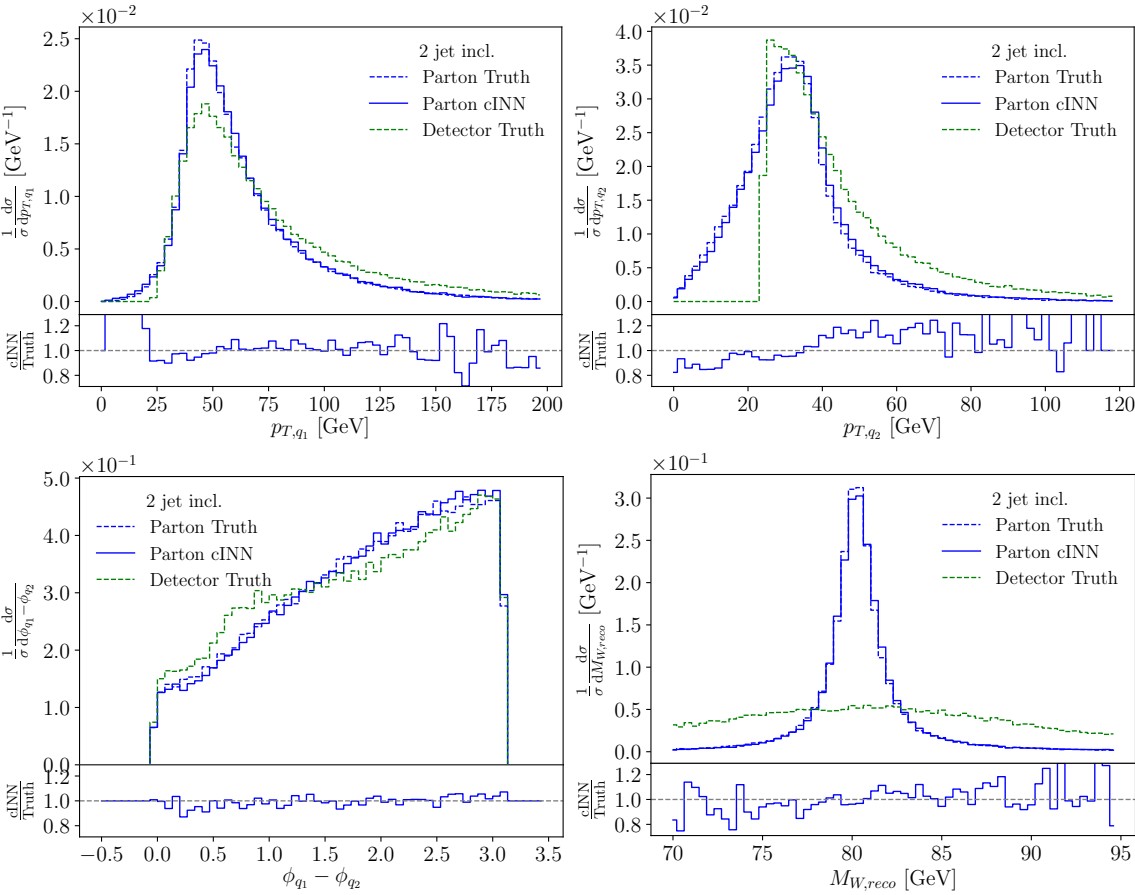

Figure 9: cINNed example distributions. Training and testing events include two to four jets, combining the samples from Fig. 6 and Fig. 8 in one network. At the parton level there exist only two $W$-decay quarks.

radiation, described either by resummed ISR or by fixed-order QCD corrections. The training sample consists of the combination of the right panels in Fig. 6 and the two panels in Fig. 8. This means that the network has to deal with the different number of jets in the final state and how they can be related to the two hard jets of the partonic $ZW \to \ell\ell jj$ process. The number of jets in the final state is not given by individual hard partons, but by the jet algorithm and its $R$-separation.

In Fig. 9 we show a set of unfolded distributions. First, we see that the $p_{T,j}$ thresholds at the detector level are corrected to allow for $p_{T,q}$ values to zero. Next, we see that the comparably flat azimuthal angle difference at the parton level is reproduced to better than 10% over the entire range. Finally, the $m_{jj}$ distribution with its MMD loss re-generates the $W$-mass peak at the parton level almost perfectly. The precision of this unfolding is not any worse than it is for the case where the number of hard partons and jets have to match and we only unfold the detector effects.

In Fig. 10 we split the unfolded distributions in Fig. 9 by the number of 2, 3, and 4 jets in the detector-level events. In the first two panels we see that the transverse momentum spectra of the hard partons are essentially independent of the QCD jet radiation. In the language of higher-order calculations this means that we can describe extra jet radiation with a constant $K$-factor, if necessary with the appropriate phase space mapping. Also the reconstruction of the $W$-mass is not affected by the extra jets, confirming that the neural network correctly identifies the $W$-decay jets and separates them from the ISR jets. Finally, we test the transverse

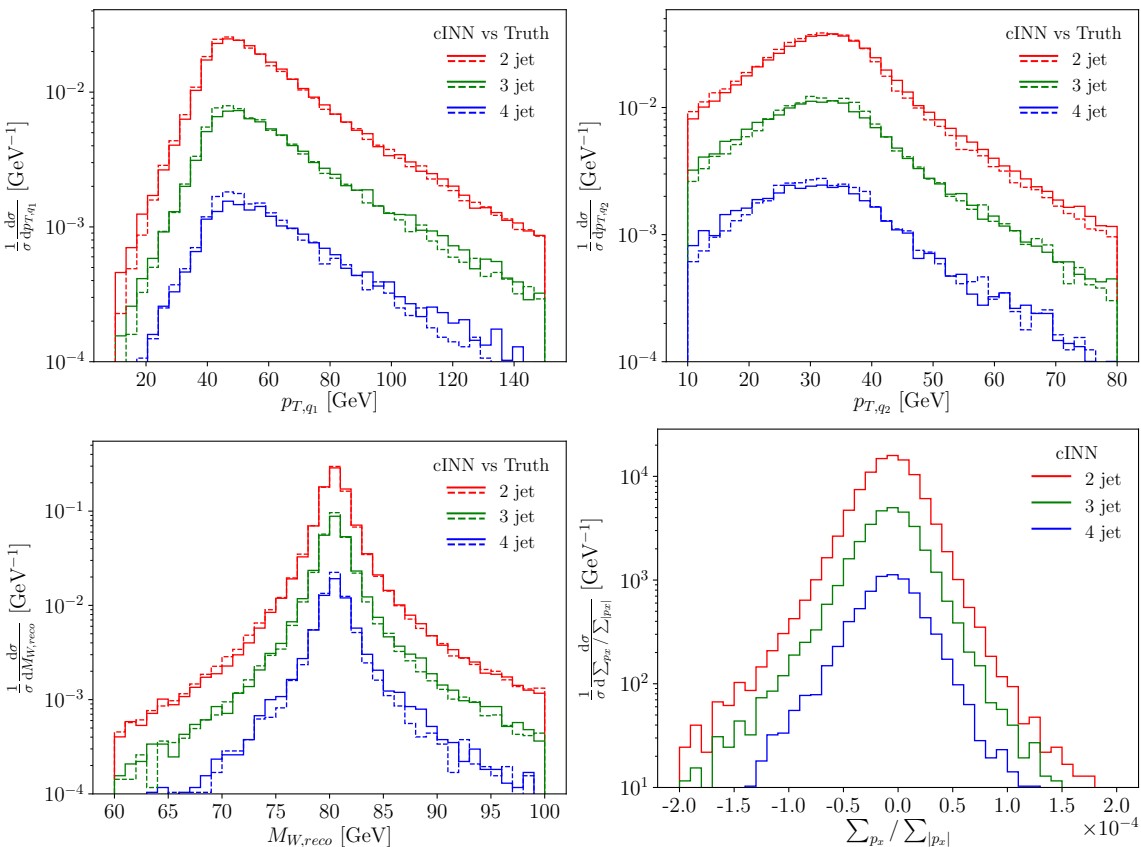

Figure 10: cINNed example distributions. Training and testing events include two to four events, combining the samples from Fig. 6 and Fig. 8 in one network. The parton-level events are stacked by number of jets at detector level.

momentum conservation at the unfolded parton level. Independent of the number of jets in the final state the energy and momentum for the pre-defined hard process is conserved at the $10^{-4}$ level. The kinematic modifications from the ISR simulation are unfolded correctly, so we can compute the matrix element for the hard process and use it for instance for inference.

## 5 Outlook

We have shown how an invertible network (INN) and in particular a conditional INN can be used to unfold detector effects for the simple example process of $ZW \to \ell\ell jj$ production at the LHC. The cINN is not only able to unfold the process over the entire phase space, it also gives correctly calibrated posterior probability distributions over parton-level phase space for given detector-level events. This feature is new even for neural network unfolding.

Next, we have extended the unfolding to a variable number of jets in the final state. This situation will automatically appear whenever we include higher-order corrections in perturbative QCD for a given hard process. The hard process at parton level is defined at the training level. We find that the cINN also unfolds QCD jet radiation in the sense that it identifies the ISR jets and corrects the kinematics of the hard process to ensure energy-momentum conservation in the hard scattering.

In combination, these features should enable analysis techniques like the matrix element method and efficient ways to communicate analysis results including multi-dimensional kine-

matic distributions. While the *ZW* production process used in this analysis, we expect these results to carry over to more complex processes with intermediate particles [21] and the impact of a SM-training hypothesis should be under control [48], the next step will be to test this new framework in a realistic LHC example with proper precision predictions and a focus on uncertainties. As for any analysis method suitable for the coming LHC runs, the challenge will be to control the full uncertainty budget at the per-cent level.[†]

# Acknowledgements

We would like to thank Ben Nachman for great discussions and Hans-Christian Schultz-Coulon for the experimental encouragement. RW and MB acknowledge support by the IMPRS-PTFS. The research of AB, MB, and TP is supported by the Deutsche Forschungsgemeinschaft (DFG, German Research Foundation) under grant 396021762 – TRR 257 *Particle Physics Phenomenology after the Higgs Discovery*. GK acknowledges support by the DFG under Germany's Excellence Strategy – EXC 2121 *Quantum Universe – 390833306*.

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
