# Peer review of "Invertible Networks or Partons to Detector and Back Again"

_SciPost Physics, doi:SciPost Phys. 9, 074 (2020)_

## Round 2 · Referee Report · Anonymous (Referee 1) · 2020-8-4

Report

In the article "Invertible Networks or Partons to Detector and Back Again" the authors present a new proof of concept that machine learning techniques can be used to dramatically improve the efficiency of collider simulations. To the best of my knowledge this is the first time in which an invertible neural network has been applied to a full simulation and therefore providing a simple way to unfold data to Parton level.

The paper is well written overall. I believe it is easily to the standard expected by SciPost but have a few remarks and questions which the authors should consider before publication.

1) Firstly, a very minor point. The subscripts 'd' and 'p' are introduced in the 2nd paragraph of section 2 but more clearly defined at the beginning of section 2.1.

3) Before equation (13) the authors state 'the form gets modified by an exponential'. I did not understand the logic here although it appears the construction has been used already in the literature. Did the authors mean 'can be modified'? If so perhaps a few more words on how this improves the numerical performance would be helpful.

4) Figures 8 and 10 do not include the useful ratio plots given in other figures. I suggest they should be added to these cases.

6) The authors may like to clarify the choice of parameters in Tables 1 and 2, and whether has been any tuning of the parameters specific to this test case. Could there be bias introduced if the 10% testing sample has also been used in the subsequent analysis?

7) The analysis of jet radiation is interesting. As I understand it this is an attempt to unfold to different stages in the Parton shower. Is this method dependent one having a leading order hard scattering before the shower in the simulation?

8) The outlook takes a very positive stance on the generalisation to any collider simulation. It would be interesting if the authors could expand upon the limitations of the current study. The reference process, pp->WZ->lljj, has a very simple resonance structure at leading order so perhaps there could be new features in other channels that would cause problems for the network. Perhaps the authors could state where they feel future developments and improvements would be interesting or necessary before the method could be reliably applied to real data.

  • validity: -
  • significance: -
  • originality: -
  • clarity: -
  • formatting: -
  • grammar: -

Author:  Ramon Winterhalder  on 2020-10-02  [id 992]

(in reply to Report 1 on 2020-08-04)

1) Firstly, a very minor point. The subscripts 'd' and 'p' are introduced in the 2nd paragraph of section 2 but more clearly defined at the beginning of section 2.1.

-> Should be more clear now.

2) Before equation (13) the authors state 'the form gets modified by an exponential'. I did not understand the logic here although it appears the construction has been used already in the literature. Did the authors mean 'can be modified'? If so perhaps a few more words on how this improves the numerical performance would be helpful.

-> We clarified this statement.

3) Figures 8 and 10 do not include the useful ratio plots given in other figures. I suggest they should be added to these cases.

-> We added the ratio plots to Figure 8. In the stacked distributions in Figure 10 we decided to omit the ratio plot for illustration purposes.

4) The authors may like to clarify the choice of parameters in Tables 1 and 2, and whether has been any tuning of the parameters specific to this test case. Could there be bias introduced if the 10% testing sample has also been used in the subsequent analysis?

-> We added a sentence about the parameter choice in Table 1 and 2.

5) The analysis of jet radiation is interesting. As I understand it this is an attempt to unfold to different stages in the Parton shower. Is this method dependent one having a leading order hard scattering before the shower in the simulation?

-> This is correct, we have improved the discussion at the beginning of Sec.4

6) The outlook takes a very positive stance on the generalisation to any collider simulation. It would be interesting if the authors could expand upon the limitations of the current study. The reference process, pp->WZ->lljj, has a very simple resonance structure at leading order so perhaps there could be new features in other channels that would cause problems for the network. Perhaps the authors could state where they feel future developments and improvements would be interesting or necessary before the method could be reliably applied to real data.

-> We have added a brief discussion to the end of the outlook, unsurprisingly we expect the main challenge to be the uncertainties.

---

## Round 2 · Referee Report · Anonymous (Referee 2) · 2020-8-13

Report

The authors discuss the use of an invertible neural network (INN) to unfold detector effects and, in addition, the QCD shower. It appears that INNs are indeed a nice tool for this purpose. The paper is well written and provided that the authors can answer satisfactorily my comments and concerns below, I would recommend it for publication.

According to the acceptance criteria of SciPost Physics (https://scipost.org/SciPostPhys/about#criteria), in my opinion this paper falls under expectation 4: 'Provide a novel and synergetic link between different research areas'. The (all required) general acceptance criteria could be met once the points below are addressed.

  1. In the second paragraph of the introduction, the authors list some shortcomings of current methods of simulation and unfolding, the two statements below appear selectively strong. 1.a They claim that one 'cannot avoid simulating events for each point in model space' when testing BSM hypotheses. This is perhaps true for some variables and only if one has a specific model to test. However, in an EFT framework where a small set of Wilson coefficients can be identified (WZ at high pT is a particular example here), a fit in the space of Wilson coefficients can be constructed avoiding the need for point by point simulation. 1.b Regarding recasting an existing analysis, the statment seems absolute while in reality this is not the case. For example see: ATL-PHYS-PUB-2020-007. Since the authors claim that these shortcomings leave us with no choice but to invert the simulation chain, I would appreciate a more objective and nuanced discussion of these points.

  2. The authors choose to invert the simulation chain back to the leading order partonic level distributions. Why is this the right choice? If the goal is to apply this inverted simulation to actual LHC events, it's far from obvious that the leading order partonic distributions are the correct choice. There are several points that enter here, first a measured distrubtion is not purely signal but also contains backgrounds which cannot be removed. In addition, some processes suffer from large QCD and even QED corrections which distort the leading order distributions. At the very least a discussion of these issues should be given and ideas as to how to address them should be profferred.

  3. If the end goal to use such a setup for bounding or discovering BSM models that contribute to diboson processes (e.g., their chosen example of WZ), where one of the electroweak bosons decays hadronically, the use of jet substructure techniques are indispensable. With this in mind, could the authors elaborate on their statement that this technique only works for analyses that don't employ jet substructure techniques?

  4. According to SciPost Physics' general acceptance criteria 3 & 5, enough details should be provided such that the results are reproducible. At least all details of the networks used should be given in an appendix in order to reproduce the results in this paper. While not required, it would be even better if sample code is shared on a git hosting platform.

  5. In general, a clear discussion of what can be gained from an event by event unfolding as they suggest is lacking.

  • validity: good
  • significance: good
  • originality: good
  • clarity: good
  • formatting: excellent
  • grammar: excellent

Author:  Ramon Winterhalder  on 2020-10-02  [id 991]

(in reply to Report 2 on 2020-08-13)

1) In the second paragraph of the introduction, the authors list some shortcomings of current methods of simulation and unfolding, the two statements below appear selectively strong.

1.a) They claim that one 'cannot avoid simulating events for each point in model space' when testing BSM hypotheses. This is perhaps true for some variables and only if one has a specific model to test. However, in an EFT framework where a small set of Wilson coefficients can be identified (WZ at high pT is a particular example here), a fit in the space of Wilson coefficients can be constructed avoiding the need for point by point simulation.

-> We are not sure we understand this criticism. We softened the statement in 
   the introduction to allow for some interpretation space, but we would argue 
   that even for SMEFT analyses in high-pT tails we need to cover parameter space. 
   Morphing techniques combining phase-space and parameter-space coverage are 
   just an especially efficient way to do this, especially when we happen 
   to know how a new-physics matrix element scales. The same is true for 
   reweighting techniques used in SMEFT-MC.

1.b) Regarding recasting an existing analysis, the statment seems absolute while in reality this is not the case. For example see: ATL-PHYS-PUB-2020-007. Since the authors claim that these shortcomings leave us with no choice but to invert the simulation chain, I would appreciate a more objective and nuanced discussion of these points.

-> We softened these statements as part of re-writing parts of the introduction.

2) The authors choose to invert the simulation chain back to the leading order partonic level distributions. Why is this the right choice? If the goal is to apply this inverted simulation to actual LHC events, it's far from obvious that the leading order partonic distributions are the correct choice. There are several points that enter here, first a measured distrubtion is not purely signal but also contains backgrounds which cannot be removed. In addition, some processes suffer from large QCD and even QED corrections which distort the leading order distributions. At the very least a discussion of these issues should be given and ideas as to how to address them should be profferred.

-> We discuss this aspect in some detail at the beginning of Sec.4. Indeed, the naive LO hard process will not always be appropriate, as we discuss using the mono-jet example. We also mention the open questions/next steps at the end of the outlook now.

3) If the end goal to use such a setup for bounding or discovering BSM models that contribute to diboson processes (e.g., their chosen example of WZ), where one of the electroweak bosons decays hadronically, the use of jet substructure techniques are indispensable. With this in mind, could the authors elaborate on their statement that this technique only works for analyses that don't employ jet substructure techniques?

-> For the detector unfolding, subjet techniques are only a matter of the complexity of the model and a availability of tracking information in the simulation, as now discussed in the introduction. For the QCD unfolding we have added a discussion to the beginning of Sec.4. Subjet techniques are fine as long as they do not interfere with our ability to define a hard process.

4) According to SciPost Physics' general acceptance criteria 3 & 5, enough details should be provided such that the results are reproducible. At least all details of the networks used should be given in an appendix in order to reproduce the results in this paper. While not required, it would be even better if sample code is shared on a git hosting platform.

-> We now state that we are using Pytorch, cite the original cINN paper with more details. Fig.5 shows the complete architecture of the cINN, and Tab.2 should include all parameters needed to code it. If any information is still missing, please let us know, we are not trying to keep information secret. Moving all this information to an appendix would, in our opinion, hurt the flow of the paper. We also added a footnote clarifying that we are very happy to share our code with interested parties. Unfortunately, our code is still very preliminary and we do not have the person power to maintain it as a public code. We also emphasize that interpreting the SciPost criteria in a constraining manner would make it impossible to publish higher-order QCD predictions or to ever attract ATLAS and CMS papers.

5) In general, a clear discussion of what can be gained from an event by event unfolding as they suggest is lacking.

-> Now included in an expanded introduction to Sec.3.

---

## Round 2 · Referee Report · Anonymous (Referee 3) · 2020-8-25

Strengths

This paper introduces for the first time the use of invertible neural networks to the key issue of unfolding. It lays the initial foundations for the development of a new generation of machine-learning based unfolding methods, which might have substantial impact on a wide range of LHC analyses.

Weaknesses

The two main weakness of this paper that I see are the following: 1. It is that it is difficult to evaluate how well the cINN algorithm performs against other state of the art ML approaches, notably OmniFold. 2. It is also very difficult to know how robust the trained unfolding algorithm is and what issues one might encounter when applying it to real data, where there is no possibility of comparing to the "truth", and the paper does not really address this issue. As such, while the ideas in the paper are promising, it is hard to evaluate if they could become competitive tools in real LHC analyses.

Report

The authors introduce a novel method using a conditional invertible neural network to invert detector simulations directly on observables. This tackles a crucial problem present when analyzing experimental data at the LHC. Overall, the paper is well written and an interesting contribution to the literature, taking a novel approach to unfolding. As such I recommend it for publication after the authors address my minor remarks.

  • In the introduction, the authors could cite 1004.2006 and 1712.01814
  • The authors mention several alternative methods, such as OmniFold, but do not provide any comparison to show the performance of their method except against their own previous FCGAN model. If a full comparison is too difficult to achieve, some discussion of the differences with 1911.09107 and 1806.00433 could be included.
  • One limitation of having a single data sample that is then split into training and test set is that it it is very difficult to know if the resulting INN suffers from overfitting. While perhaps imperfect, one could probe this by applying a (conditional) INN trained on a specific process, tune, or parton shower on a sample generated differently, and gain at least some sense of the robustness of the unfolding method.
  • It would be helpful to include a ratio of cINN to truth in figures 8 and 10 as was done in other figures.
  • The paper makes broad claims about the applicability of the method to high-level observables, however only a handful of observables are shown. Have the authors looked at others, e.g. jet shapes or substructure observables?

Requested changes

See report

  • validity: good
  • significance: high
  • originality: high
  • clarity: top
  • formatting: excellent
  • grammar: excellent

Author:  Ramon Winterhalder  on 2020-10-02  [id 990]

(in reply to Report 3 on 2020-08-25)
Category:
answer to question

1) In the introduction, the authors could cite 1004.2006 and 1712.01814

-> Done

2) The authors mention several alternative methods, such as OmniFold, but do not provide any comparison to show the performance of their method except against their own previous FCGAN model. If a full comparison is too difficult to achieve, some discussion of the differences with 1911.09107 and 1806.00433 could be included.

-> The case of the original GAN proposal (1806.00433) is covered by the comparison with the FCGAN, where by construction the pure GAN will do worse. For Omnifold we could only speculate. We added a footnote on p.15 to the point that we indeed need a community study on the different approaches eventually.

3) One limitation of having a single data sample that is then split into training and test set is that it it is very difficult to know if the resulting INN suffers from overfitting. While perhaps imperfect, one could probe this by applying a (conditional) INN trained on a specific process, tune, or parton shower on a sample generated differently, and gain at least some sense of the robustness of the unfolding method.

-> As in FCGAN with W' -> into text

4) It would be helpful to include a ratio of cINN to truth in figures 8 and 10 as was done in other figures.

-> We added the ratio plots to Figure 8. In the stacked distributions in Figure 10 we decided to omit the ratio plot for illustration purposes.

5) The paper makes broad claims about the applicability of the method to high-level observables, however only a handful of observables are shown. Have the authors looked at others, e.g. jet shapes or substructure observables?

-> We have looked at all observables describing the hard process and decided to show the most interesting. More distributions are hidden in the multi-page pdf files included in the arXiv source code, as now stated in the caption to Fig.3. At this stage we do not train on sub-jet observables, so they are currently not available.

---

## Round 3 · Referee Report · Anonymous (Referee 3) · 2020-10-20

Report

The authors have addressed my main comments and I recommend the article for publication in SciPost

---

## Round 3 · Referee Report · Anonymous (Referee 1) · 2020-11-3

Report

In the revised version of the manuscript the authors have addressed comments of all the reviewers. I recommend the paper for publication.

---

## Round 3 · Referee Report · Anonymous (Referee 2) · 2020-11-3

Report

The authors have satisfactorily addressed my comments and I recommend the manuscript for publication.

---

## Editorial Decision

published